# Risk factors and economic burden of postoperative anastomotic leakage related events in patients who underwent surgeries for colorectal cancer

**Jeonghyun Kang[1], Hyesung Kim[2], HyeJin Park[2], Bora Lee[3,4], Kang Young Lee[5]***

**1** Department of Surgery, Gangnam Severance Hospital, Yonsei University College of Medicine, Seoul, Republic of Korea, **2** HEMA, Johnson & Johnson Medical Korea, Seoul, Republic of Korea, **3** Institute of Health & Environment, Seoul National University, Seoul, Korea, **4** RexSoft Corporation, Seoul, South Korea, **5** Department of Surgery, Severance Hospital, Yonsei University College of Medicine, Seoul, Republic of Korea

* kylee117@yuhs.ac

**Data Availability Statement:** Under the public, single-payer system, the claim data were generated in the Health Insurance Review and Assessment Service (HIRA) archives in the process of

## Abstract

### Background

Nationwide research about the clinical and economic burden caused by anastomotic leakage (AL) has not been published yet in Korea. This study assessed the AL rate and quantified the economic burden using the nationwide database.

### Methods

This real world evidence study used health claims data provided by the Korean Health Insurance Review and Assessment Service (HIRA, which showed that 156,545 patients underwent anterior resection (AR), low anterior resection (LAR), or ultra-low anterior resection (uLAR) for colorectal cancer (CRC) between January 1, 2007 and January 31, 2020. The incidence of AL was identified using a composite operational definition, a composite of imaging study, antibacterial drug use, reoperation, or image-guided percutaneous drainage. Total hospital costs and length of stay (LOS) were evaluated in patients with AL versus those without AL during index hospitalization and within 30 days after the surgery.

### Results

Among 120,245 patients who met the eligibility criteria, 7,194 (5.98%) patients had AL within 30 days after surgery. Male gender, comorbidities (diabetes, metastatic disease, ischemic heart disease, ischemic stroke), protective ostomy, and multiple linear stapler use, blood transfusion, and urinary tract injury were associated with the higher odds of AL. Older age, rectosigmoid junction cancer, AR, LAR, and laparoscopic approach were related with the reduced odds of AL. Patients with AL incurred higher costs for index hospitalization compared to those without AL (8,991 vs. 7,153 USD; p<0.0001). Patients with AL also required

reimbursing healthcare providers, covering all Korean citizens. In order to use the big data of HIRA, a researcher should submit the application with a study proposal. HIRA review the application for approval of the access to data. Once the approval is given, HIRA retrieves the data from the data warehouse system, which is then uploaded to the system of HIRA. The system is accessible only to the researcher for the study through the designated computer of the HIRA datacenter or authorized remote access only for a limited period. As taking the raw data out is forbidden, the researcher can take out only the analysis results. Therefore, we cannot share the minimal data set of this study. The application process and documents are specified in the Healthcare Bigdata Hub homepage of HIRA. HIRA homepage address: https://opendata.hira.or.kr/or/orb/useGdInfo.do.

**Funding:** This work was supported by Johnson & Johnson Medical Korea(https://www.jnjmedicaldevices.com/ko-KR). HK and HJP, employees of the sponsor, participated in study design and preparation of the manuscript. BL who was paid for the statistical analysis from the sponsor. The funder provided support in the form of salaries for authors, HK, HJP, and BL, but did not play any additional role in the study design, data collection and analysis, decision to publish it, or preparation of the manuscript. The specific roles of these authors are articulated in the 'author contributions' section.

**Competing interests:** The authors have read the journal's policy and have the following competing interests to declare. HK and HJP are employees of Johnson & Johnson Medical Korea. BL is an employee of RexSoft, which was paid for the statistical analysis from Johnson & Johnson Medical Korea. There are no patents, products in development or marketed products associated with this research to declare. This does not alter our adherence to PLOS ONE policies on sharing data and materials.

longer LOS (16.78 vs. 14.22 days; p<0.0001) and readmissions (20.83 vs. 13.93 days; p<0.0001).

## Conclusion

Among patients requiring resection for CRC, the occurrence of AL was associated with significantly increased costs and LOS. Preventing AL could not only produce superior clinical outcomes, but also reduce the economic burden for patients and payers.

## Introduction

Colorectal cancer (CRC) was the fourth most common cancer following lung, breast, and prostate cancer, and was the second leading cause of cancer related deaths around the world in 2018 [1]. According to the annual report of cancer statistics in Korea in 2017, CRC is the second most common cancer type, with 28,111 new cases and the third leading cause of cancer-related deaths behind lung and liver cancer [2]. Mutiple surgical approaches, including hemicolectomy (right or left), anterior resection, and low anterior resection, are commonly indicated for patients with CRC [3].

Complications of colorectal resection can be divided into intraoperative and postoperative ones. Intraoperative complications include bleeding, bowel injuries, ureteral lesions, and bladder injuries. The most frequent postoperative complications include surgical site infection, anastomotic leakage (AL), intraabdominal abscess, ileus, and bleeding [4]. Among these, AL is considered to be the most detrimental in terms of the impact on morbidity, mortality and quality of life [5–8]. AL leads to anastomotic stricture and impaired colorectal function including reduced neorectal capacity, evacuation problems, fecal urgency, and incontinence [9, 10]. More seriously, AL is associated with an increase in local recurrence and lower long-term survival [11–14]. Alongside these complications and the potential need for reoperation, AL management consumes extensive healthcare resources and expenses [8, 15–17].

The rate of AL has been reported to range from 5% to 19% globally in colorectal/coloanal surgery depending upon perioperative factors such as anatomical site, surgical method, operators' experience, and patient's clinical characteristics [6, 18]. Although a multicenter study reported an AL rate of 6.3% after laparoscopic rectal cancer surgery in Korea, the AL rate at each hospital showed wide variation ranging from 2.0% to 10.3% [19]. To identify the generalizable economic burden caused by AL, a nationwide population study is necessary but has not been carried out yet in Korea.

This retrospective, nationwide claims analysis was conducted to identify clinical and economic outcomes among Korean patients with cancer who had undergone anterior resection (AR), low anterior resection (LAR), or ultra-low anterior resection (uLAR) with a manual circular stapler as the current standard of care. This study sought to assess an association between such factors as patients, procedures, and providers and the incidence of AL, and its economic burden in terms of length of stay (LOS), readmissions, and total costs.

## Methods

### Data sources

Under the public and single-payer system, health claims data have been generated in the archives of the Health Insurance Review and Assessment Service (HIRA) in the process of

reimbursing healthcare providers, covering all Korean citizens. As these reimbursements are predominantly based on the fee-for-service system, claims data contain comprehensive information about treatments, pharmaceuticals, procedures, and diagnoses for almost 50 million beneficiaries. This study was approved by the Public Institutional Bioethics Committee designated by the Ministry of Health & Welfare (MOHW) (No. P01-202007-21-024) including the waiver of informed consent as it was a retrospective study based on a de-identified claims database.

## Study population

The HIRA database identified 156,545 patients who underwent AR, LAR, or uLAR (procedure codes: Q2921, QA921, Q2922, QA922, Q2928, QA928) between January 1, 2007 and January 31, 2020, of whom only adult patients (≥19 years) who had undergone surgery as the principal treatment for CRC (ICD-10 code: C18, C19, C20) using manual circular staplers (device code: B1022XXX) were included. After excluding patients who had prior colorectal surgery within a 1-year lookback period, the analysis dataset included a total of 120,245 patients. The "index surgery" was defined as the first colorectal resection for each patient, and the "index hospitalization" as admission for the index surgery.

## Clinical outcomes

The following variables were identified as the indicators of surgical complications within 30 days after the index surgery: AL, blood transfusion, urinary tract injury, pulmonary embolism, acute renal failure.

Since there was no specific diagnosis code for AL, the following procedures were required for AL occurrence during in-hospital care to meet operational AL definitions: (1) Imaging study including computed tomography scans (2) Administration of antibacterial drugs (more than 7 consecutive days after the surgery) (3) Abdominal reoperation and (4) image-guided percutaneous drainage. AL occurrence is defined by the combination of the above four items. Combinations to meet the definition of AL are any including both (1) and (2), any including (3), or any including (4). Among these combinations to define AL, those that include (3) or (4) were designated as additional intervention cases (AIC), because reoperation and percutaneous drainage are potentially more definitive treatments for AL compared to imaging and antibiotics.

Blood transfusion was detected using the procedure code. Urinary tract injury, pulmonary embolism, and acute renal failure were all also identified through the ICD-10 codes.

## Economic outcomes

The economic outcomes identified were as follows: LOS for the index hospitalization, LOS for all-cause readmissions within 30 days, total costs for the index hospitalization, total costs for during the period from the index surgery to 30 days post-operation, and total costs for readmissions. To comprehensively understand the economic burden on society, total costs included both out-of-pocket expenditures for patients and costs covered by payer.

## Covariates

The following covariates of interest were included: demographics (age, gender), patient characteristics (primary diagnosis for the index surgery, Charlson Comorbidity Index (CCI), other comorbidities (diabetes, chronic obstructive pulmonary disease (COPD), metastatic disease, ischemic heart disease, ischemic stroke), pre-operation radiation therapy, procedure

characteristics (surgical procedure, surgical approach, protective ostomy, multiple circular stapler use, multiple linear stapler use), provider characteristics (number of beds) and others (blood transfusion, urinary tract injury).

## Statistical analysis

All study variables were analyzed descriptively. Continuous variables were presented as mean, median, and standard deviation, while categorical variables were expressed as frequencies and proportions.

The overall incidence of each complication (AL, additional intervention cases, blood transfusion, urinary tract injury, pulmonary embolism, acute renal failure) was identified, and the incidence of AL was further summarized by surgical procedure (AR, LAR, or uLAR), primary diagnosis (C18, C19, C20) and surgical approach (open or laparoscopic).

To figure out the factors associated with AL, all covariates considered in this study were compared according to the incidence of AL. Student's t-test or Wilcoxon's rank sum test was applied for comparison of continuous variables while a chi-squared test or Fisher's exact test was conducted for categorical variables as appropriate after testing assumptions such as normality. The univariable and multivariable logistic regression models were fitted to assess the risk factors associated with AL so that the odds ratio of each risk factor could be calculated. The multivariable model included factors that showed the statistical significance based on univariable analysis. LOS and costs were estimated during a 30-day period after the index surgery. Costs were summarized by surgical procedure and approach and compared with the Kruskal-Wallis (K-W) test. When the costs difference was significant by the K-W test, then the Dwass, Steel, Critchlow-Fligner method was applied for post hoc comparison.

All analyses were conducted using SAS (version 9.2, SAS Institutes, Cary, NC, USA) and R (version 4.0.3, The R Foundation for statistical computing, Vienna, Austria). A p-value of 0.05 or less was considered to indicate statistically significant difference.

## Ethical considerations

The study was reviewed by the Public Institutional Review Board designated by the Ministry of Health and Welfare and determined to be exempt from IRB approval (Review number: P01-202007-21-024). As the study involves no more than minimal risk to patients, IRB approved a request to waive informed consent under the Bioethics and Safety Act of Korea.

## Results

### Incidence of anastomotic leakage and additional interventional cases based on the operational definition

Among 156,545 patients who underwent AR, LAR, or uLAR from January 2007 to January 2020, a total of 120,245 patients met the eligibility criteria (Fig 1) for this study. Among these patients, 33.97% had imaging studies, 7.54% received antibacterial drugs for more than 7 consecutive days, 0.36% had reoperations, and 6.99% had image-guided percutaneous drainage (Table 1). 5.98% of patients satisfied the operational definition of AL having events corresponding with the defined combination within 30 days after surgery. When applying a stricter definition that requires reoperation or image-guided percutaneous drainage, 4.56% of patients fulfilled the definition of additional interventional cases (AIC).

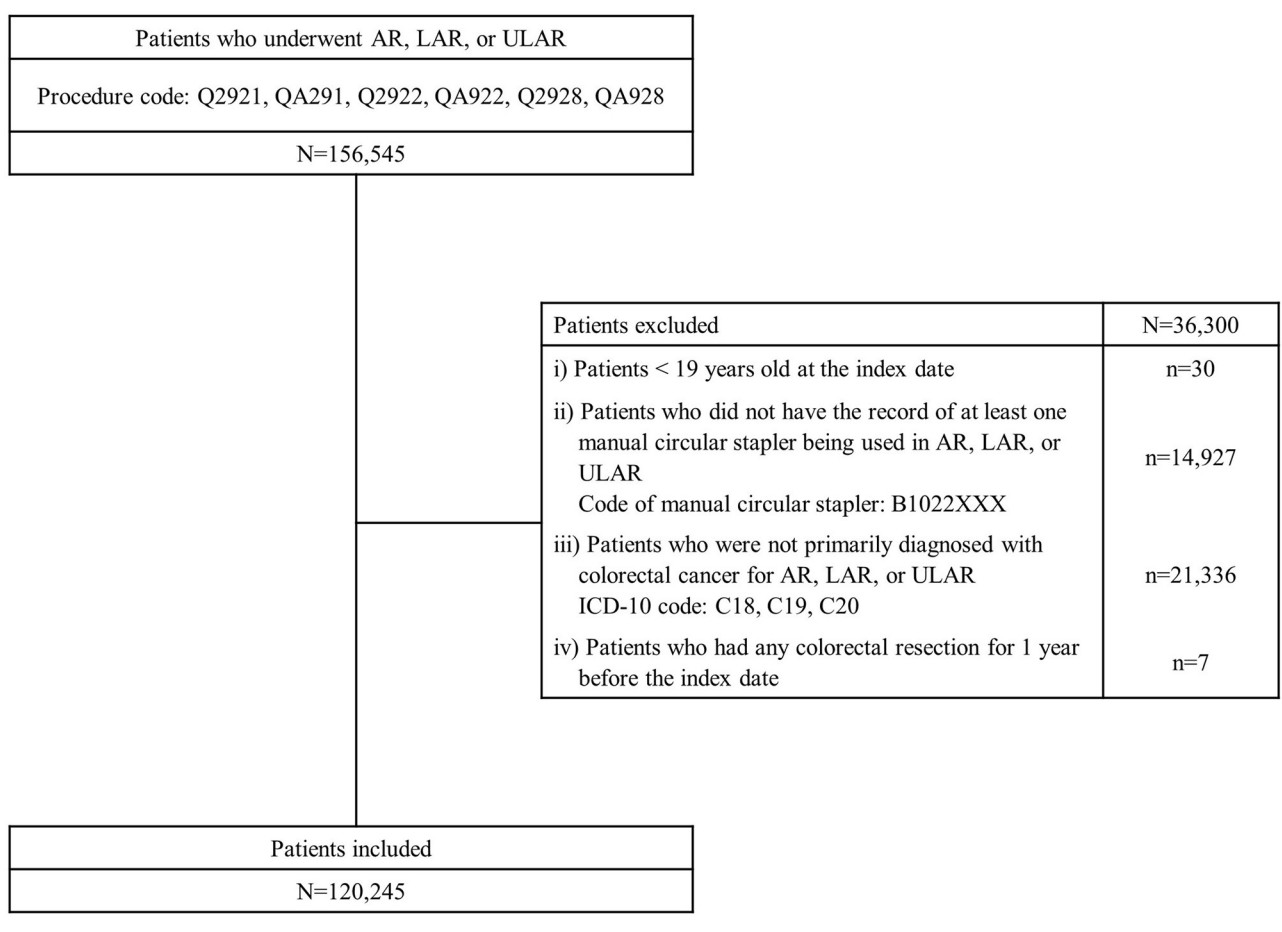

| Patients who underwent AR, LAR, or ULAR | |
|---|---|
| Procedure code: Q2921, QA291, Q2922, QA922, Q2928, QA928 | |
| N=156,545 | |

| Patients excluded | N=36,300 |
|---|---|
| i) Patients < 19 years old at the index date | n=30 |
| ii) Patients who did not have the record of at least one manual circular stapler being used in AR, LAR, or ULAR<br>Code of manual circular stapler: B1022XXX | n=14,927 |
| iii) Patients who were not primarily diagnosed with colorectal cancer for AR, LAR, or ULAR<br>ICD-10 code: C18, C19, C20 | n=21,336 |
| iv) Patients who had any colorectal resection for 1 year before the index date | n=7 |

| Patients included | |
|---|---|
| N=120,245 | |

**Fig 1. Pateint selection flow chart.**

## Demographics and perioperative clinical characteristics

Patient demographics and perioperative characteristics are displayed in Table 2. Their mean age was 64.09 (standard deviation [SD] ± 11.45) years, and there was no statistically significant difference between patients with and without AL. The proportion of males was significantly higher in patients with AL (66.68% vs. 62.41%, $p<0.0001$). More patients with AL had LAR or uLAR than those without AL (LAR: 61.44% vs. 53.57% and uLAR: 2.93% and 1.61%, $p<0.0001$). Patients with AL had more open surgery (32.50% vs. 25.47%, $p<0.0001$), protective ostomy (27.15% vs. 15.83%, $p<0.0001$), multiple linear stapler use (57.83% vs. 48.94%, $p<0.0001$), multiple circular stapler use (1.08% vs. 0.76%, $p = 0.0027$), and blood transfusion (32.44% vs. 17.00%, $p<0.0001$) than those without AL. The following comorbidities were more frequently observed in patients with AL than those without AL; diabetes (24.48% vs. 20.62%, $p<0.0001$), COPD (7.17% vs. 6.33%, $p = 0.0046$), metastatic disease (12.65% vs. 7.41%, $p<0.0001$), ischemic heart disease (10.19% vs. 8.22%, $p<0.0001$), and ischemic stroke (5.35% vs. 3.97%, $p<0.0001$). The CCI score in the AL group was significantly higher than that in patients without AL (mean ± SD, 3.75 ± 2.90 vs. 3.18 ± 2.43, $p<0.0001$).

## Risk factors for anastomotic leakage

The results from logistic regression predicting the odds of AL are shown in Table 3. Most of the variables had an association with AL in univariable analysis, except for age and the number

**Table 1. Anastomotic leakage and other clinical outcomes.**

| | Total | |
|---|---|---|
| | (N = 120,245) | |
| Item 1 (Imaging Study) | 40,851 | (33.97%) |
| Item 2 (Administration of antibacterial drug) | 9,068 | (7.54%) |
| Item 3 (Abdominal reoperation) | 427 | (0.36%) |
| Item 4 (Image guided percutaneous drainage) | 8,406 | (6.99%) |
| Anastomotic leakage (within 30 days) | | |
| No | 113,051 | (94.02%) |
| Yes | 7,194 | (5.98%) |
| Combination to meet the definition of AL | | |
| Combination 1 (all two items of 1 and 2 without 3 or 4) | 1,711 | (23.78%)† |
| Combination 2 (any combination including 3 without 4) | 40 | (0.56%)† |
| Combination 3 (any combination including 4 without 3) | 5,291 | (73.55%)† |
| Combination 4 (any combination including 3 and 4) | 152 | (2.11%)† |
| Additional intervention cases (within 30 days) | | |
| No | 114,762 | (95.44%) |
| Yes | 5,483 | (4.56%) |
| Combination to meet the definition of AIC | | |
| Combination 2 (any combination including 3 without 4) | 40 | (0.73%)†† |
| Combination 3 (any combination including 4 without 3) | 5,291 | (96.50%)†† |
| Combination 4 (any combination including 3 and 4) | 152 | (2.77%)†† |

†Denominator = Number of patients with anastomotic leakage (7,194)

††Denominator = Number of patients with additional intervention cases (5,483).

of hospital beds. In multivariable analysis, male gender had the increased odds of AL (odds ratio [OR]: 1.176, 95% confidence interval [CI]: 1.117 to 1.239), and older age was associated with the reduced odds of AL (OR 0.995, 95% CI, 0.993 to 0.997). A primary diagnosis of C19 (rectosigmoid junction cancer) decreased the odds of AL (OR 0.888, 95% CI, 0.821 to 0.96, reference: C20 (rectal cancer)), but C18 (colon cancer) did not have a statistically significant association with AL. Diabetes (OR 1.124, 95% CI, 1.059 to 1.192), metastatic disease (OR 1.44, 95% CI, 1.335 to 1.553), ischemic heart disease (OR 1.156, 95% CI, 1.064 to 1.257), and ischemic stroke (OR 1.237, 95% CI 1.107 to 1.382) each increased the odds of AL. Among operative characteristics, protective ostomy (OR 1.546, 95% CI 1.448 to 1.65) and multiple linear stapler use (OR 1.192, 95% CI 1.133 to 1.255) were associated with the higher odds of AL. Meanwhile, AR and LAR were associated with the lower odds of AL versus uLAR (OR 0.619, 95% CI 0.524 to 0.731 and OR 0.691, 95% CI 0.594 to 0.805), and the laparoscopic approach was also associated with the lower odds of AL (OR 0.849, 95% CI 0.804 to 0.897).

## Economic outcomes

We compared the total healthcare costs (the index hospitalization alone, the period from the index date to last follow-up date, and readmissions) by procedure and approach (Table 4) (Fig 2). All cost categories showed the highest costs in patients who received uLAR compared to AR and LAR. For the index hospitalization, uLAR with an open approach had the highest mean cost (11,212 ± 6,198 USD), and AR with a laparoscopic approach had the lowest mean cost (6,689 ± 3,256 USD). While open approaches were more costly than laparoscopic

**Table 2. Demographics and perioperative clinical characteristics.**

| | With AL | | Without AL | | Total | | P-value |
|---|---|---|---|---|---|---|---|
| | (N = 7,194) | | (N = 113,051) | | (N = 120,245) | | |
| Age, (years) | | | | | | | 0.1347 |
| Mean ± SD | 63.89 | ± 11.67 | 64.10 | ± 11.43 | 64.09 | ± 11.45 | |
| Median (IQR) | 64.00 | (56.00, 73.00) | 65.00 | (56.00, 73.00) | 65.00 | (56.00, 73.00) | |
| Gender, n(%) | | | | | | | <0.0001 |
| Male | 4,797 | (66.68) | 70,553 | (62.41) | 75,350 | (62.66) | |
| Female | 2,397 | (33.32) | 42,498 | (37.59) | 44,895 | (37.34) | |
| Primary diagnosis | | | | | | | <0.0001 |
| C18 (colon) | 2,801 | (38.94) | 51,587 | (45.63) | 54,388 | (45.23) | |
| C19 (colorectal junction) | 1,007 | (14.00) | 17,917 | (15.85) | 18,924 | (15.74) | |
| C20 (rectum) | 3,386 | (47.06) | 43,547 | (38.52) | 46,933 | (39.03) | |
| CCI Score | | | | | | | |
| Mean ± SD | 3.75 | ± 2.90 | 3.18 | ± 2.43 | 3.21 | ± 2.46 | <0.0001 |
| Median | 3.00 | (2.00, 5.00) | 2.00 | (2.00, 4.00) | 3.00 | (2.00, 4.00) | |
| Comorbidities at baseline | | | | | | | |
| Diabetes | 1,761 | (24.48) | 23,315 | (20.62) | 25,076 | (20.85) | <0.0001 |
| COPD | 516 | (7.17) | 7,156 | (6.33) | 7,672 | (6.38) | 0.0046 |
| Metastatic disease | 910 | (12.65) | 8,372 | (7.41) | 9,282 | (7.72) | <0.0001 |
| Ischemic heart disease | 733 | (10.19) | 9,292 | (8.22) | 10,025 | (8.34) | <0.0001 |
| Ischemic stroke | 385 | (5.35) | 4,488 | (3.97) | 4,873 | (4.05) | <0.0001 |
| Radiation therapy | 922 | (12.82) | 9,845 | (8.71) | 10,767 | (8.95) | <0.0001 |
| Surgical procedure | | | | | | | <0.0001 |
| AR | 2,563 | (35.63) | 50,669 | (44.82) | 53,232 | (44.27) | |
| LAR | 4,420 | (61.44) | 60,560 | (53.57) | 64,980 | (54.04) | |
| uLAR | 211 | (2.93) | 1,822 | (1.61) | 2,033 | (1.69) | |
| Surgical approach | | | | | | | <0.0001 |
| Open | 2,338 | (32.50) | 28,789 | (25.47) | 31,127 | (25.89) | |
| Laparoscopic | 4,856 | (67.50) | 84,262 | (74.53) | 89,118 | (74.11) | |
| Protective ostomy | 1,953 | (27.15) | 17,894 | (15.83) | 19,847 | (16.51) | <0.0001 |
| Multiple circular stapler use | 78 | (1.08) | 863 | (0.76) | 941 | (0.78) | 0.0027 |
| Multiple linear stapler use (≥3) | 4,160 | (57.83) | 55,327 | (48.94) | 59,487 | (49.47) | <0.0001 |
| Year of surgery | | | | | | | <0.0001 |
| 2008~2011 | 2,148 | (29.86) | 35,398 | (31.31) | 37,546 | (31.22) | |
| 2012~2015 | 2,355 | (32.74) | 38,605 | (34.15) | 40,960 | (34.06) | |
| 2016~2020 | 2,691 | (37.41) | 39,048 | (34.54) | 41,739 | (34.71) | |
| Number of beds | | | | | | | 0.1745 |
| 0~499 beds | 995 | (13.83) | 16,330 | (14.44) | 17,325 | (14.41) | |
| 500~999 beds | 3,692 | (51.32) | 56,863 | (50.30) | 60,555 | (50.36) | |
| 1000 or more | 2,507 | (34.85) | 39,858 | (35.26) | 42,365 | (35.23) | |
| Blood transfusion (within 30 days) | 2,334 | (32.44) | 19,219 | (17.00) | 21,553 | (17.92) | <0.0001 |
| Urinary tract injury | 18 | (0.25) | 86 | (0.08) | 104 | (0.09) | <0.0001 |
| Pulmonary embolism | 45 | (0.63) | 152 | (0.13) | 197 | (0.16) | <0.0001 |
| Acute renal failure | 190 | (2.64) | 254 | (0.22) | 444 | (0.37) | <0.0001 |

AL = anastomotic leakage; CCI = Charlson comorbidity index; SD = standard deviation; COPD = chronic obstructive pulmonary disease; AR = anterior resection;

LAR = low anterior resection; uLAR = ultra-low anterior resection

Data is presented as the number of patients (%) except for age and CCI score and the denominator of % is the total number of patients in each category.

**Table 3. Risk factors and clinical parameters associated with the anastomotic leak from logistic regression.**

| Variable | | Univariable | | Multivariable | |
|---|---|---|---|---|---|
| | | Odds Ratio (95% CI) | p-value | Odds Ratio (95% CI) | p-value |
| Age, (years) | | 0.998 (0.996, 1.000) | 0.1268 | 0.995 (0.993, 0.997) | <.0001 |
| Gender | Male vs Female | 1.205 (1.146, 1.268) | <.0001 | 1.176 (1.117, 1.239) | <.0001 |
| Primary diagnosis | C18 vs C20 | 0.698 (0.663, 0.735) | <.0001 | 0.936 (0.868, 1.009) | 0.0851 |
| | C19 vs C20 | 0.723 (0.672, 0.777) | <.0001 | 0.888 (0.821, 0.960) | **0.0028** |
| CCI Score | | 1.086 (1.077, 1.096) | <.0001 | - | |
| Diabetes (at baseline) | Yes vs No | 1.248 (1.180, 1.319) | <.0001 | 1.124 (1.059, 1.192) | **0.0001** |
| COPD (at baseline) | Yes vs No | 1.143 (1.042, 1.255) | **0.0046** | 1.087 (0.988, 1.195) | 0.0869 |
| Metastatic disease (at baseline) | Yes vs No | 1.811 (1.683, 1.948) | <.0001 | 1.44 (1.335, 1.553) | <.0001 |
| Ischemic heart disease (at baseline) | Yes vs No | 1.267 (1.171, 1.372) | <.0001 | 1.156 (1.064, 1.257) | **0.0007** |
| Ischemic stroke (at baseline) | Yes vs No | 1.369 (1.230, 1.523) | <.0001 | 1.237 (1.107, 1.382) | **0.0002** |
| Radiation therapy | Yes vs No | 1.541 (1.434, 1.656) | <.0001 | 1.014 (0.933, 1.103) | 0.7432 |
| Surgical procedure | AR vs uLAR | 0.437 (0.377, 0.506) | <.0001 | 0.619 (0.524, 0.731) | <.0001 |
| | LAR vs uLAR | 0.630 (0.545, 0.729) | <.0001 | 0.691 (0.594, 0.805) | <.0001 |
| Surgical approach | Laparoscopic vs Open | 0.709 (0.674, 0.747) | <.0001 | 0.849 (0.804, 0.897) | <.0001 |
| Protective ostomy | Yes vs No | 1.982 (1.877, 2.092) | <.0001 | 1.546 (1.448, 1.650) | <.0001 |
| Multiple circular stapler use | Yes vs No | 1.427 (1.131, 1.801) | **0.0028** | 1.196 (0.944, 1.514) | 0.1377 |
| Multiple linear stapler use (≥3) | Yes vs No | 1.430 (1.363, 1.501) | <.0001 | 1.192 (1.133, 1.255) | <.0001 |
| Number of beds | 0~499 vs 1000 or more | 0.969 (0.898, 1.045) | 0.4104 | - | |
| | 500~999 vs 1000 or more | 1.032 (0.980, 1.088) | 0.2341 | - | |
| Blood transfusion | | 2.345 (2.227, 2.469) | <.0001 | 2.039 (1.928, 2.156) | <.0001 |
| Urinary tract injury | | 3.295 (1.981, 5.479) | <.0001 | 2.606 (1.549, 4.384) | **0.0003** |

The covariates of multiple model were selected based on a significance level of 0.15 in the univariable model.

CCI and comorbidities (diabetes, COPD, metastatic disease, ischemic heart disease, ischemic stroke) had high multicollinearity, and the multiple model with comorbidities showed better explanatory power than that with CCI.

approaches for the index hospitalization for both AR and uLAR, an open approach for LAR was associated with lower index hospitalization costs than the laparoscopic approach.

## Economic burden of anastomotic leakage

Economic outcomes for patients with AL and without AL are summarized in Table 5. The mean costs for the index hospitalization were significantly higher for patients with AL compared to those without AL (8,991 vs. 7,153 USD; $p<0.0001$). Including the 30-day follow-up period, the mean costs were 10,971 USD for patients with AL and 7,531 USD for those without AL ($p<0.0001$), reflective of a marked increase in post-discharge resource use among patients in the AL cohort. Patients with AL also required prolonged LOS for the index hospitalization versus those without AL (16.78 days vs. 14.22 days; $p<0.0001$). Readmission costs for patients with AL were higher than those for patients without AL (3,160 vs. 1,316 USD; $p<0.0001$). AL patients also required longer mean LOS for readmissions compared to those without AL (20.83 days vs. 13.93 days; $p<0.0001$). The S1 Table shows economic outcomes depending on the presence of AIC.

## Discussion

This study demonstrates that AL after surgery for colorectal cancer is associated with increased costs for the index hospitalization and readmissions. this study is one of the largest studies to

**Table 4. Healthcare costs by procedure & approach.**

| Variable | | Surgical procedure | | | p-value | p-value by post hoc comparison* | | |
|---|---|---|---|---|---|---|---|---|
| | | AR | LAR | uLAR | by KW test | AR vs. LAR | AR vs. uLAR | LAR vs. uLAR |
| Total costs for the index hospitalization (USD) | | | | | <.0001 | | | |
| Open | N | 11,534 | 19,332 | 261 | | <.0001 | <.0001 | <.0001 |
| | Mean ± SD | 6,726 ± 4,409 | 7,069 ± 4,598 | 11,212 ± 6,198 | | | | |
| | Median (IQR) | 5,593 (4,257–7,756) | 5,969 (4,678–7,951) | 9,324 (8,127–12,636) | | | | |
| Laparoscopic | N | 41,698 | 45,648 | 1,772 | | <.0001 | <.0001 | <.0001 |
| | Mean ± SD | 6,689 ± 3,256 | 7,842 ± 3,719 | 10,893 ± 3,574 | | | | |
| | Median (IQR) | 6,124 (5,047–7,434) | 7,121 (5,863–8,693) | 9,986 (8,944–11,676) | | | | |
| p-value by post hoc comparison* | | <.0001 | <.0001 | 0.0251 | | | | |
| Total cost for the period from index date to last f/u date (USD) | | | | | <.0001 | | | |
| Open | N | 11,534 | 19,332 | 261 | | <.0001 | <.0001 | <.0001 |
| | Mean ± SD | 7,216 ± 4,535 | 7,526 ± 4,723 | 11,954 ± 6,218 | | | | |
| | Median (IQR) | 6,131 (4,665–8,314) | 6,408 (5,028–8,523) | 10,077 (8,559–13,902) | | | | |
| Laparoscopic | N | 41,698 | 45,648 | 1,772 | | <.0001 | <.0001 | <.0001 |
| | Mean ± SD | 7,151 ± 3,423 | 8,318 ± 3,869 | 11,629 ± 4,030 | | | | |
| | Median (IQR) | 6,559 (5,414–7,971) | 7,475 (6,229–9,279) | 10,684 (9,438–12,673) | | | | |
| p-value by post hoc comparison* | | <.0001 | <.0001 | 0.1957 | | | | |
| Total cost for the readmission (USD) | | | | | <.0001 | | | |
| Open | N | 2,328 | 3,536 | 64 | | 0.2464 | 0.0382 | <.0001 |
| | Mean ± SD | 1,532 ± 2,174 | 1,679 ± 2,359 | 2,137 ± 2,078 | | | | |
| | Median (IQR) | 1,293 (755–1,917) | 1,111 (635–1,848) | 1,430 (1,090–2,947) | | | | |
| Laparoscopic | N | 8,113 | 9,216 | 399 | | <.0001 | <.0001 | <.0001 |
| | Mean ± SD | 1,618 ± 1,792 | 1,603 ± 1,910 | 2,325 ± 2,542 | | | | |
| | Median (IQR) | 1,344 (811–1,853) | 1,142 (639–1,851) | 1,579 (970–2,590) | | | | |
| p-value by post hoc comparison* | | 0.9473 | 0.9932 | 0.9972 | | | | |

1 USD = 1,150 Korean won; USD, U.S. dollar

KW = Kruskal-Wallis test; AR = anterior resection; LAR = low anterior resection; uLAR = ultra-low anterior resection; SD = standard deviation; IQR = interquartile range

* Post hoc pairwise multiple comparison analysis was performed by the Dwass, Steel, Critchlow-Fligner method.

measure the economic burden of AL after colorectal surgery, and using a nationwide population dataset, it could overcome selection bias inherent to smaller, single institution studies.

Various clinical parameters were significant predictors of AL in our analysis. Male gender, diabetes mellitus, ischemic heart disease, ischemic stroke, uLAR, multiple stapler usage, blood transfusion, and urinary tract injury are reported to be well-known risk factors for a higher rate of AL [20–23]. However, there were several associations that were a matter of controversy. Previous prospective randomized clinical trials for rectal cancer showed no significant difference in the incidence of AL between open and laparoscopic surgeries [24–26]. In contrast, one cohort study revealed an increased rate of AL in laparoscopic surgery versus open surgery (10.8% vs. 3.4%, p = 0.012) in patients with mid and low rectal cancer [27]. This result, however, needs to be interpreted with caution, and the learning curve inexperience in the introduction period of laparoscopic surgery for rectal cancer may affect the higher AL rate. Our study showed that the odds of AL were statistically lower for laparoscopic surgery than for open

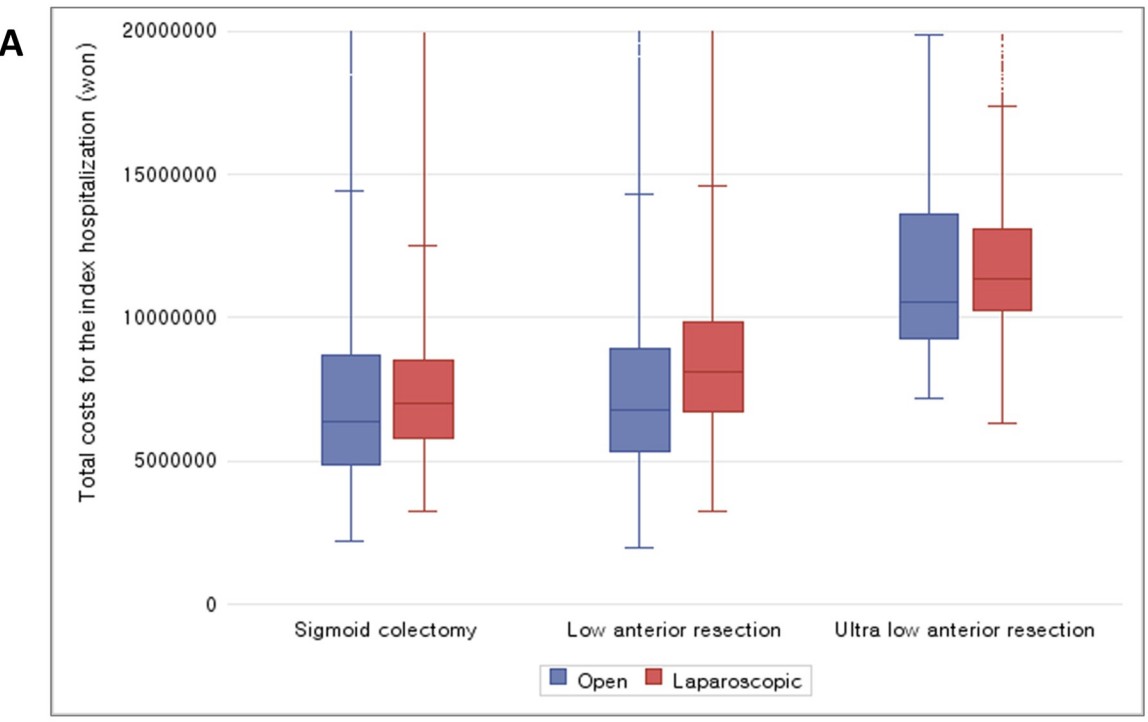

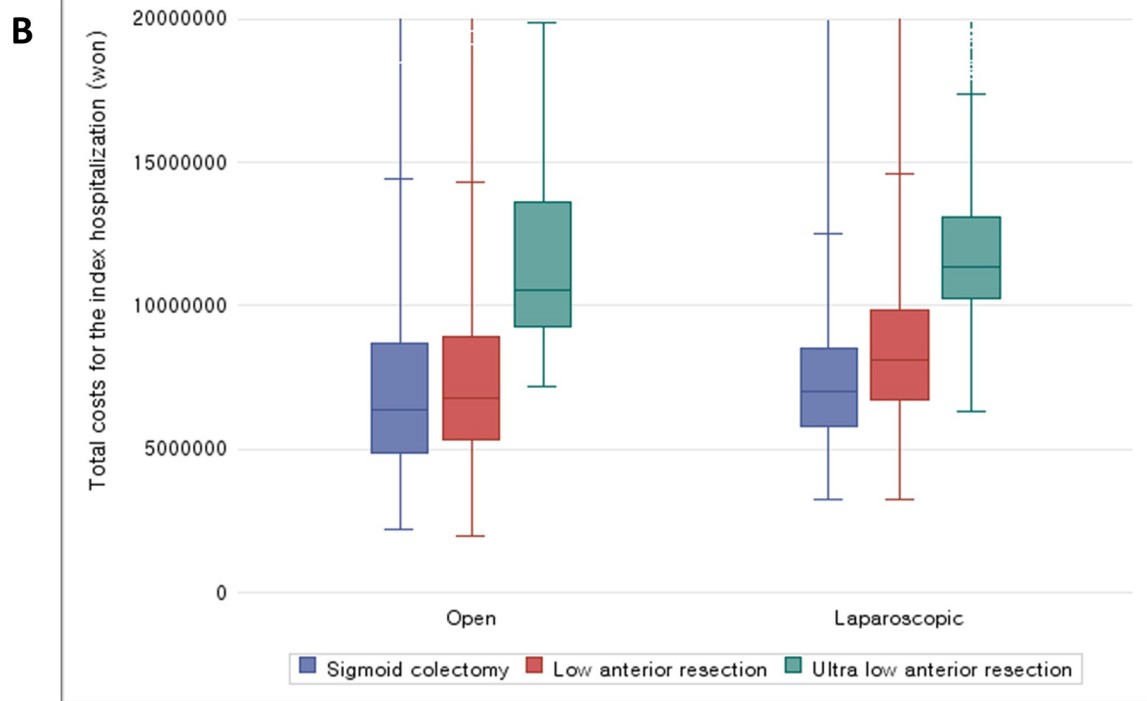

**Fig 2. Healthcare costs by procedure & approach.** The clipped boxplot trimmed extremely high values based on 20,000,000 KRW. (1 U.S. dollor = 1,150 Korean won) (A) The number of clipped patients in open anterior resection = 256; The number of clipped patients in laparoscopic anterior resection = 334; The number of clipped patients in open low anterior resection = 432; (B) The number of clipped patients in laparoscopic low anterior resection = 792; The number of clipped patients in open ultra-low anterior resection = 17; The number of clipped patients in laparoscopic ultra-low anterior resection = 76.

**Table 5. Economic outcomes of the presence of anastomotic leakage.**

| | With AL | Without AL | Total | P-value |
|---|---|---|---|---|
| | (N = 7,194) | (N = 113,051) | (N = 120,245) | |
| Total costs for the index hospitalization, (USD) | | | | <0.0001 |
| Mean ± SD | 8,991 ± 6,415 | 7,153 ± 3,619 | 7,263 ± 3,868 | |
| Median | 7,399 | 6,457 | 6,508 | |
| Q1, Q3 | 5,782–9,965 | 5,213–8,101 | 5,239–8,186 | |
| Total costs for the period from the index date to last F/U date, (USD) | | | | <0.0001 |
| Mean ± SD | 10,971 ± 6,977 | 7,531 ± 3656 | 7,737 ± 4,018 | |
| Median | 9,222 | 6,851 | 6,950 | |
| Q1, Q3 | 7,138–12,505 | 5,538–8,565 | 5,592–8,743 | |
| Total costs for readmissions, (USD) | | | | <0.0001 |
| N | 5,762 | 51,260 | 57,022 | |
| Mean ± SD | 3,160 ± 3,815 | 1,316 ± 1,029 | 1,643 ± 1,987 | |
| Median | 2,000 | 1,134 | 1,241 | |
| Q1, Q3 | 1,117–3,762 | 655–1,703 | 705–1,866 | |
| LOS for the index hospitalization, (duration) | | | | <0.0001 |
| Mean ± SD | 16.78 ± 13.24 | 14.22 ± 8.36 | 14.37 ± 8.75 | |
| Median | 13.00 | 12.00 | 12.00 | |
| Q1, Q3 | 10.00–19.00 | 10.00–16.00 | 10.00–16.00 | |
| LOS for readmissions, (duration) | | | | <0.0001 |
| N | 5,762 | 51,260 | 57,022 | |
| Mean ± SD | 20.83 ± 16.61 | 13.93 ± 10.58 | 14.63 ± 11.53 | |
| Median | 17.00 | 13.00 | 13.00 | |
| Q1, Q3 | 10.00–27.00 | 7.00–18.00 | 7.00–18.00 | |

1 USD = 1,150 Korean won; USD, U.S. dollar

AL = anastomotic leakage; SD = standard deviation.

surgery. Basically, there are inevitably significant differences in patient selection, surgical procedure, and complication risk between the patient group who underwent laparoscopic surgery and the group who underwent open surgery. A decision on whether to select laparoscopic or open surgery necessarily involves various preoperative evaluations and surgeon's concerns about anastomotic leakage risks. In addition, the selection of a surgical method can play a role in preventing the occurrence of anastomotic leakage, but it was very difficult to verify this point in a retrospective study using claim data. Another important confounding factor may be a conversion from laparoscopic to open surgery, which could be an indicator of difficulties encountered during surgery. During such conversions anastomosis may be performed manually, which may help to prevent AL. Unfortunately, conversion surgeries could not be ascertained using claims data, so our findings with respect to surgical approach and incidence of AL should be interpreted with caution.

The influence of preoperative chemoradiotherapy (preop-CRT), especially for rectal cancer, on the incidence of AL is also up for discussion. Jang et al. reported that preop-CRT did not have any incremental impact on the occurrence of AL, as the incidence of AL in patients with and without preop-CRT was 7.5% and 8.1%, respectively, after propensity matching ($p$ = 0.781) [28]. Hu et al. reported that neoadjuvant therapy did not statistically increase the incidence of AL (OR 1.16, 95% CI 0.99–1.36; $p$ = 0.07, random effects model) in a meta-analysis [29]. However, other reports revealed that preop-CRT was a significant independent factor

for AL [19, 30]. Our large cohort study showed preoperative receipt of radiation therapy to have no statistical association with the odds of AL. Further studies are required to elucidate the true impact of preop-CRT with AL.

With respect to the influence of age on the occurrence of AL, Zaimi et al. reported that increased age every 5 years was protective for AL after colorectal resections in the multivariable analysis (OR 0.965, 95% CI 0.941–0.985, $p<0.001$) of 45,488 Dutch colorectal cancer patients [31]. Similarly, other population-based studies revealed that old age was associated with a lower risk of AL [20, 32]. In contrast, several studies did not support this or demonstrated that old age was a risk factor for a higher AL rate [19, 33–35]. In this study, we found that age was associated with a lower incidence of AL. Surgical intervention performed by surgeons with caution for elderly patients or healthy survivor effect, indicating that unhealthy patients die before reaching older age might be suggested as possible reasons for less AL in elderly patients [31]. Our study is one of the largest studies that demonstrated the protective effect of age on AL in Asian groups. Although our study did not focus on the impact of age on AL, this observation might be helpful in planning adequate surgical procedures for elderly patients.

A previous study found short-term outcomes to be dependent on hospital caseload and a degree of specialization [36]. Zheng et al. demonstrated this hospital-center effect with respect to hospital LOS and in-hospital mortality for patients with stage I-III colon cancer who were treated with laparoscopic colectomies [37]. Among the 120,245 patients who were enrolled in our study, only 17,325 (14.41%) underwent surgeries in hospitals with a small number of beds (less than 500 beds). This phenomenon might be attributed to the concentration of hospitals selected by patients in Korea, as more patients from rural areas tend to travel to large metropolitan hospitals for surgery. We did not detect differences in the odds of AL as a function of the hospital size (over 1000 beds, 500–1000 beds, and less than 500 beds). Therefore, surgical processes and/or population risks may not vary across hospitals within these size categories. Curative resection for CRC is regarded as a complex procedure which requires highly trained and experienced surgeons. It was not possible to conduct the analysis of this factor, since the source data lack information on hospital and surgeon caseload.

Hospital LOS was longer among AL patients than those without AL in our study, though the gap between the two groups was relatively small (mean 16.78 days in the AL group vs. 14.22 days in the no AL group, $p<0.001$). In contrast, Lee et al. observed a much larger incremental difference in the USA (12 days in the AL group vs. 5 days in the no AL group, $p<0.0001$) [17]. In Korea, many patients may not seek early discharge, given that hospital room charges are reimbursed by the national insurance system. The impact of AL on hospital LOS may therefore depend on historical practices and incentives within local healthcare systems. Despite modest differences in LOS, the total costs for the index hospitalization were significantly higher in patients with AL, so were the total costs for readmissions; these observations corroborate findings from a previous study [17].

One of the critical limitations of this study is that the presence of AL was not identified using individual patients' medical records. The characteristics of claim data archived by the Korean Health Insurance Review and Assessment Service (HIRA) are not suitable to find a definitive case of AL. There is a precedent, however, of using surrogate indicators of AL from similarly structured data sources. Ashraf et al. reported the economic impact of AL using 23,388 patients registered at the Hospital Episode Statistics (HES) dataset from various National Health Service (NHS) hospitals in England [38]. In that study, researchers defined the AL event as "relaparotomy within 28 days of surgery" because no valid diagnostic code for AL was available [38]. In a recent study using a large claims dataset in the United States, Lee et al. inferred the occurrence of AL from the presence of an abscess, septicemia, peritonitis, or infection among 239,350 patients undergoing colorectal surgery [17]. As with these prior

studies, we selected possible clinical patterns representing ALs according an "operational definition". Nonetheless, we could not make definitive conclusions about the presence of AL, leading us to define additional interventional cases (AIC) in order to overcome the shortcomings. Repeated analyses using the definition of AIC confirmed that the relevant indicators and economic aspects were quite similar. However, as a prior study noted, not all of the relaparotomy within 28 days was done due to AL only [38]; in the case of our study, surgical relaparotomy and radiologic intervention were not always performed on account of AL only.

Another major limitation of this study relates to a lack of cancer staging data. Surgical extent, combined resection, and receipt of chemotherapy are largely dependent on cancer stage. Thus, a stage-specific comparison would be required to eliminate this confounding factor. Since claims data lack staging information, further research with a data source inclusive of surgical morbidity, leakage status, and cancer stage is required.

In conclusion, this research demonstrated AL to be associated with a significantly increased economic burden in the nationwide dataset. Despite limitations inherent to the reliance on surrogate endpoints to find AL cases, this work underscores an opportunity to reduce the economic burden associated with treatments required for AL.

## Supporting information

**S1 Table. Economic outcomes of the presence of Additional Intervention Cases(AIC).** (DOCX)

## Author Contributions

**Conceptualization:** Jeonghyun Kang, Hyesung Kim, HyeJin Park, Kang Young Lee.

**Formal analysis:** Bora Lee.

**Project administration:** Hyesung Kim, HyeJin Park.

**Writing – original draft:** Jeonghyun Kang, Hyesung Kim.

**Writing – review & editing:** HyeJin Park, Kang Young Lee.

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
