## [Decision Letter · Decision Letter 0]

18 Nov 2021

PONE-D-21-34976Risk factors and economic burden of postoperative anastomotic leakage related events in patients who underwent surgeries for colorectal cancerPLOS ONE Dear Dr. Lee,

Thank you for submitting your manuscript to PLOS ONE. After careful consideration, we feel that it has merit but does not fully meet PLOS ONE’s publication criteria as it currently stands. Therefore, we invite you to submit a revised version of the manuscript that addresses the points raised during the review process.

As you can see, the article needs improvement, especially, in the light of methodology of how the findings compare colon to rectal cancer and surgery with neoadjuvant treatment.

A marked-up copy of your manuscript that highlights changes made to the original version. You should upload this as a separate file labeled 'Revised Manuscript with Track Changes'.An unmarked version of your revised paper without tracked changes. You should upload this as a separate file labeled 'Manuscript'.

We look forward to receiving your revised manuscript.

Kind regards,

Alberto Meyer, MD, PhD

Academic Editor

PLOS ONE

Journal Requirements:

(This work was supported by Johnson & Johnson Medical Korea(https://www.jnjmedicaldevices.com/ko-KR). HK and HJP, employees of the sponsor, participated in study design and preparation of the manuscript. BL who was paid for the statistical analysis from the sponsor.).

3. We note that one or more of the authors is affiliated with the funding organization, indicating the funder may have had some role in the design, data collection, analysis or preparation of your manuscript for publication; in other words, the funder played an indirect role through the participation of the co-authors. If the funding organization did not play a role in the study design, data collection and analysis, decision to publish, or preparation of the manuscript and only provided financial support in the form of authors' salaries and/or research materials, please do the following:

A. Review your statements relating to the author contributions, and ensure you have specifically and accurately indicated the role(s) that these authors had in your study. These amendments should be made in the online form.

B. Confirm in your cover letter that you agree with the following statement, and we will change the online submission form on your behalf:

“The funder provided support in the form of salaries for authors [insert relevant initials], but did not have any additional role in the study design, data collection and analysis, decision to publish, or preparation of the manuscript. The specific roles of these authors are articulated in the ‘author contributions’ section.

C. Please note that in order to use the direct billing option the corresponding author must be affiliated with the chosen institute. Please either amend your manuscript to change the affiliation or corresponding author, or email us at plosone@plos.org with a request to remove this option.

4.Thank you for stating the following in the Competing Interests section: 

(I have read the journal's policy and the authors of this manuscript have the following competing interests: HK and HJP are employees of Johnson & Johnson Medical Korea. BL is an employee of RexSoft, which was paid for the statistical analysis from Johnson & Johnson Medical Korea.). 

Additional Editor Comments:

After careful consideration, we feel that it has merit but does not fully meet PLOS ONE’s publication criteria as it currently stands.

Therefore, we invite you to submit a revised version of the manuscript that addresses the points raised during the review process. As you can see, the article needs improvement, especially, in the light of methodology of how the findings compare colon to rectal cancer and surgery with neoadjuvant treatment.

We look forward to receiving your revised manuscript.

King regards,

Alberto Meyer

Academic Editor

PLOS ONE

Reviewers' comments:

Reviewer's Responses to Questions

**Comments to the Author**

1. Is the manuscript technically sound, and do the data support the conclusions?

Reviewer #1: Yes

Reviewer #2: Yes

Reviewer #3: Yes

2. Has the statistical analysis been performed appropriately and rigorously? 

Reviewer #1: I Don't Know

Reviewer #2: Yes

Reviewer #3: Yes

3. Have the authors made all data underlying the findings in their manuscript fully available?

Reviewer #1: Yes

Reviewer #2: Yes

Reviewer #3: Yes

4. Is the manuscript presented in an intelligible fashion and written in standard English?

Reviewer #1: Yes

Reviewer #2: Yes

Reviewer #3: Yes

5. Review Comments to the Author

Reviewer #1: The English language needs to be polished for correcting some writing and alphabetical errors thorough text.

It is suggested that statistical analyses are reviewed/re-checked by a statistician.

The statistical values and numbers are suggested to be reviewed again to confirm by authors.

Reviewer #2: The authors retrospectively analyzed the risk factors and economic burden of anastomatic leakage (AL) after colorectal cancer. The study included large samples from Korean patient database, the analysis method was proper, and the manuscript was well organized. However, the limitations of the study were also obvious. There are some important issues to be further addressed before acceptance:

1. The study include rectal cancer and sigmoid colon cancer, which were highly heterogeneous in surgery and the AL risk. Why did the author mixed such two kinds of disease together?

2. In the method part, the author clarified that "all patients underwent surgery as the primary treatment", that means all the included patients did not receive neoadjuvant chemoradiotherapy. So the conclusion of this study is not representative to all the CRC patients who got surgery, but only to those who got surgery first (without neoadjuvant chemoradiotherapy).

3. The AL has classification criteria of AL (grade), which was helpful to evaluate the severity of AL, and also helpful to explain the cost and LOS data, but the authors did not show the grade data.

4. Due to the big difference in therapy, the result has its own bias and limitation. For example, the open surgery and laparoscopy surgery has huge diffence in patient selection, surgical procedure and complication risk; so the authors need fully discuss the limitation of the results.

Reviewer #3: Authors conducted a retrospective, nationwide research about the clinical and economic burden caused by anastomotic leakage (AL) in Korea. Of 156,545 patients undergoing anterior resection (AR), low anterior resection (LAR), or ultra-low anterior resection (uLAR) for colorectal cancer (CRC) between January 1, 2007 and January 31, 2020 were included. Among 120,245 patients who met the eligibility criteria, 7,194 (5.98%) patients had AL within 30 days after surgery. Male gender, comorbidities, protective ostomy, and multiple linear stapler use were associated with a higher odds of AL. Older age, rectosigmoid junction cancer, AR, LAR, and laparoscopic approach were associated with reduced odds of AL. Patients with AL incurred higher costs for index hospitalization compared to those without AL (8,991 vs. 7,153 USD; p <0.0001). Patients with AL also required longer LOS (16.78 vs. 14.22 days; p <0.0001) and readmissions (20.83 vs. 13.93 days; p <0.0001). In summary, they concluded that patients requiring resection for CRC, the occurrence of AL was associated with significantly increased costs and LOS. Preventing AL could not only provide for superior clinical outcomes, but also reduce the economic burden for patients and payers. The results seems interesting and appealing; however, there are a lot of criticisms and have several issues that the authors need to address before the manuscript is suitable for publication.

Major Compulsory Revisions:

1. Clinical outcomes paragraph. The following variables were identified as indicators of surgical complications within 30 days after index surgery: AL, infection, blood transfusion, urinary tract injury, ileus, pneumonia, pulmonary embolism, acute renal failure. Blood transfusion is defined as surgical complications? The transfusion units should be considered. Acute myocardial infarction, wound infection and stroke should be also included as surgical complications.

2. Since there is no specific diagnosis code for AL, presence of the following procedures was required when AL occurrence during in-hospital care comprised operational AL definitions: (1) Imaging study including computed tomography scans (2) Administration of antibacterial drugs (more than 7 consecutive days after the surgery), the above two procedures were hard to be defined as AL. In addition, the nonsynchronous creation of colostomy after AR, LAR, and laparoscopic approach should be considered as AL.

3. In Table 2: Demographics and Perioperative Clinical characteristics. How did authors could identify some variables were surgical complications or risk factors of AL? For example, ischemic heart disease, ischemic stroke, etc.

4. Table 3: Risk factors of anastomotic leak from logistic regression. Age, (years) was an independent variable by multivariate analysis but not by univariate analysis? The subgroup analysis of CCI Score should be mentioned here, and no multivariate analysis for CCI score? Protective ostomy is often performed in ultra-low anterior resection (uLAR), of which might be considered as a compounding factor but not as a risk factor. Furthermore, robotic-assisted surgery vs laparoscopy vs open surgery is suggested to be analyzed if this procedure is related to AL?

5. Table 4. Healthcare costs by procedures & approaches. Only open vs laparoscopic approaches? How about in the comparison with robotic-assisted surgery? The relevant information regarding robotic-assisted surgery is important in recent years.

6. Table 5. Economic outcomes by the presence of anastomotic leakage. Mean LOS and Median LOS for index hospitalization, (duration) was 14.22 ± 8.36 and 12 days, respectively. In fact, it was relatively longer compared to Western countries and even longer than some Asian countries.

7. If authors could use Health Insurance Review and Assessment Service (HIRA) archives claim data to analyze the difference of overall survival between AL vs non-AL patients?

8. In Abstract section: Male gender, comorbidities, protective ostomy, and multiple linear stapler use were associated with a higher odds of AL. The above statement should be amended according to complete results in Table 3.

Minor Essential Revisions:

1. Please correct the typos and grammatical error by English-editing with the certificate enclosed.

2. Abbreviations in the Tables must be shown their corresponding full name in the footnotes.

6. PLOS authors have the option to publish the peer review history of their article (what does this mean?). If published, this will include your full peer review and any attached files.

Reviewer #1: No

Reviewer #2: No

Reviewer #3: No

---

## [Author Response · Author response to Decision Letter 0]

31 Mar 2022

Thank you for submitting your manuscript to PLOS ONE. After careful consideration, we feel that it has merit but does not fully meet PLOS ONE’s publication criteria as it currently stands. Therefore, we invite you to submit a revised version of the manuscript that addresses the points raised during the review process.

As you can see, the article needs improvement, especially, in the light of methodology of how the findings compare colon to rectal cancer and surgery with neoadjuvant treatment.

Journal Requirements:

(This work was supported by Johnson & Johnson Medical Korea(https://www.jnjmedicaldevices.com/ko-KR). HK and HJP, employees of the sponsor, participated in study design and preparation of the manuscript. BL who was paid for the statistical analysis from the sponsor.).

3. We note that one or more of the authors is affiliated with the funding organization, indicating the funder may have had some role in the design, data collection, analysis or preparation of your manuscript for publication; in other words, the funder played an indirect role through the participation of the co-authors. If the funding organization did not play a role in the study design, data collection and analysis, decision to publish, or preparation of the manuscript and only provided financial support in the form of authors' salaries and/or research materials, please do the following:

A. Review your statements relating to the author contributions, and ensure you have specifically and accurately indicated the role(s) that these authors had in your study. These amendments should be made in the online form.

Response: Thank you for your comments. We reviewed the author contribution and modified it. This statement was applied in the revised manuscript and online form.

B. Confirm in your cover letter that you agree with the following statement, and we will change the online submission form on your behalf:

“The funder provided support in the form of salaries for authors [insert relevant initials], but did not have any additional role in the study design, data collection and analysis, decision to publish, or preparation of the manuscript. The specific roles of these authors are articulated in the ‘author contributions’ section.

Response: Thank you for your comments. We agree with your suggestion to revise the statement because the funders did not have any additional role in the study and manuscript. 

C. Please note that in order to use the direct billing option the corresponding author must be affiliated with the chosen institute. Please either amend your manuscript to change the affiliation or corresponding author, or email us at plosone@plos.org with a request to remove this option.

4.Thank you for stating the following in the Competing Interests section: 

(I have read the journal's policy and the authors of this manuscript have the following competing interests: HK and HJP are employees of Johnson & Johnson Medical Korea. BL is an employee of RexSoft, which was paid for the statistical analysis from Johnson & Johnson Medical Korea.). 

Response: We used the claim data generated in the Health Insurance Review and Assessment Service (HIRA). HIRA has strict regulations to provide the data for research. We described the legal restrictions to sharing the minimal data set in the cover letter. 

Response: We used the claim data generated in the Health Insurance Review and Assessment Service (HIRA). HIRA has strict regulations to provide the data for research. We described the legal restrictions to sharing the minimal data set in the cover letter. 

<Legal restrictions to sharing the minimal data set>

Under the public, single-payer system, the claim data were generated in the Health Insurance Review and Assessment Service (HIRA) archives in the process of reimbursing healthcare providers, covering all Korean citizens. In order to use the big data of HIRA, a researcher should submit the application with a study proposal. HIRA review the application for approval of the access to data. Once the approval is given, HIRA retrieves the data from the data warehouse system, which is then uploaded to the system of HIRA. The system is accessible only to the researcher for the study through the designated computer of the HIRA datacenter or authorized remote access only for a limited period. As taking the raw data out is forbidden, the researcher can take out only the analysis results. Therefore, we cannot share the minimal data set of this study. The application process and documents are specified in the Healthcare Bigdata Hub homepage of HIRA.

HIRA homepage address: https://opendata.hira.or.kr/or/orb/useGdInfo.do

Response: Thank you for your detailed explanation. 

Reviewers' comments:

Reviewer's Responses to Questions

Comments to the Author

1. Is the manuscript technically sound, and do the data support the conclusions?

Reviewer #1: Yes

Reviewer #2: Yes

Reviewer #3: Yes

 2. Has the statistical analysis been performed appropriately and rigorously?

 Reviewer #1: I Don't Know

Reviewer #2: Yes

Reviewer #3: Yes

 3. Have the authors made all data underlying the findings in their manuscript fully available?

 Reviewer #1: Yes

Reviewer #2: Yes

Reviewer #3: Yes

 4. Is the manuscript presented in an intelligible fashion and written in standard English?

 Reviewer #1: Yes

Reviewer #2: Yes

Reviewer #3: Yes

5. Review Comments to the Author

Reviewer #1: The English language needs to be polished for correcting some writing and alphabetical errors thorough text.

Response: Thank you for your comments. We made corrections to the manuscript accordingly. 

It is suggested that statistical analyses are reviewed/re-checked by a statistician.

Response: As you commented, the statistician reviewed/re-checked the overall statistical analysis parts. 

The statistical values and numbers are suggested to be reviewed again to confirm by authors.

Response: We reviewed all the statistical values and numbers again.

Reviewer #2: The authors retrospectively analyzed the risk factors and economic burden of anastomatic leakage (AL) after colorectal cancer. The study included large samples from Korean patient database, the analysis method was proper, and the manuscript was well organized. However, the limitations of the study were also obvious. There are some important issues to be further addressed before acceptance:

1. The study include rectal cancer and sigmoid colon cancer, which were highly heterogeneous in surgery and the AL risk. Why did the author mixed such two kinds of disease together?

Response: Basically, we agree that colon and rectal cancer have different anastomotic leakage rates. Therefore, there have been many previous studies where the analyses were performed separately. Our study was conducted with the intention of identifying the trend of anastomotic leakage for all colorectal cancer patients nationwide. As you pointed out, we analyzed the differences in anastomotic leakage rates by location such as the colon and the rectum as shown in Table 2. We also presented the healthcare costs by tumor location in Table 4. Thank you for your comments.

2. In the method part, the author clarified that "all patients underwent surgery as the primary treatment", that means all the included patients did not receive neoadjuvant chemoradiotherapy. So the conclusion of this study is not representative to all the CRC patients who got surgery, but only to those who got surgery first (without neoadjuvant chemoradiotherapy).

Response: Thank you for your comments. The sentence "all patients underwent surgery as the primary treatment” was meant to include all the patients who underwent surgeries. As you pointed out, however, this may cause misunderstandings about patients who received neoadjuvant treatment. Therefore, it would be better to modify the above expression to "all patients underwent surgery as the principal treatment" as follows. Thank you for your comments. 

3. The AL has classification criteria of AL (grade), which was helpful to evaluate the severity of AL, and also helpful to explain the cost and LOS data, but the authors did not show the grade data.

Response: As you commented, hospital costs and length of stay vary depending on the grade of AL, so if we can know this accurately, we can perform more detailed analysis. However, it was difficult to obtain data on the grades of AL due to the nature of the claim data, so we could not include it in our analysis. Thank you for your comments.

4. Due to the big difference in therapy, the result has its own bias and limitation. For example, the open surgery and laparoscopy surgery has huge diffence in patient selection, surgical procedure and complication risk; so the authors need fully discuss the limitation of the results.

Response: Thank you for your comments. We agree with your comments. We added the paragraph below in the Discussion section 

Included in the Discussion section 

Basically, there are inevitably significant differences in patient selection, surgical procedure, and complication risk between the patient group who underwent laparoscopic surgery and the group who underwent open surgery. A decision on whether to select laparoscopic or open surgery necessarily involves various preoperative evaluations and surgeon's concerns about anastomotic leakage risks. In addition, the selection of a surgical method can play a role in preventing the occurrence of anastomotic leakage, but it was very difficult to verify this point in a retrospective study using claim data.

Reviewer #3: Authors conducted a retrospective, nationwide research about the clinical and economic burden caused by anastomotic leakage (AL) in Korea. Of 156,545 patients undergoing anterior resection (AR), low anterior resection (LAR), or ultra-low anterior resection (uLAR) for colorectal cancer (CRC) between January 1, 2007 and January 31, 2020 were included. Among 120,245 patients who met the eligibility criteria, 7,194 (5.98%) patients had AL within 30 days after surgery. Male gender, comorbidities, protective ostomy, and multiple linear stapler use were associated with a higher odds of AL. Older age, rectosigmoid junction cancer, AR, LAR, and laparoscopic approach were associated with reduced odds of AL. Patients with AL incurred higher costs for index hospitalization compared to those without AL (8,991 vs. 7,153 USD; p <0.0001). Patients with AL also required longer LOS (16.78 vs. 14.22 days; p <0.0001) and readmissions (20.83 vs. 13.93 days; p <0.0001). In summary, they concluded that patients requiring resection for CRC, the occurrence of AL was associated with significantly increased costs and LOS. Preventing AL could not only provide for superior clinical outcomes, but also reduce the economic burden for patients and payers. The results seems interesting and appealing; however, there are a lot of criticisms and have several issues that the authors need to address before the manuscript is suitable for publication.

Major Compulsory Revisions:

1. Clinical outcomes paragraph. The following variables were identified as indicators of surgical complications within 30 days after index surgery: AL, infection, blood transfusion, urinary tract injury, ileus, pneumonia, pulmonary embolism, acute renal failure. Blood transfusion is defined as surgical complications? The transfusion units should be considered. Acute myocardial infarction, wound infection and stroke should be also included as surgical complications.

Response: Thank you for your comments. Blood transfusion was included as one of the variables in Table 3. Since the situation requiring blood transfusion may reflect the difficulty of surgery, we checked the relationship between blood transfusion and anastomotic leakage. However, like your opinion, there are parts that it is not easy to refer to as risk factors directly, so we have changed the title of Table 3 into "Risk factors and clinical parameters associated with the anastomotic leak from logistic regression." 

However, it was very difficult to include the transfusion volume, acute myocardial infarction, wound infection and stroke as covariates due to the characteristics of claim data. This would be one of the main limitations of our study. 

2. Since there is no specific diagnosis code for AL, presence of the following procedures was required when AL occurrence during in-hospital care comprised operational AL definitions: (1) Imaging study including computed tomography scans (2) Administration of antibacterial drugs (more than 7 consecutive days after the surgery), the above two procedures were hard to be defined as AL. In addition, the nonsynchronous creation of colostomy after AR, LAR, and laparoscopic approach should be considered as AL.

Response: We agree with your comments. We believe that abdominal CT is performed again during the period of discharge after surgery when it is really necessary to identify a certain serious situation in the abdomen. Therefore, we determined that such examination after colorectal cancer surgery would be required mostly when complications such as leakage are suspected. In addition, if antibiotics are used for more than a week, in Korea, the use of antibiotics is usually limited to 1-2 days after elective colorectal cancer surgery, according to the evaluation criteria of the National Review and Assessment Service. This standard was published as an index that most hospitals satisfy. Therefore, it can be judged that the clinical situation in which antibiotics are used continuously for one week or more after surgery is an emergency in which the use of antibiotics is essential in light of these indicators. For this reason, two items were included in the operational definition. Nevertheless, as we described the limitations in defining the accurate anastomotic leakage using nationwide claim data, it was impossible to collect the correct cases only. 

Because nonsynchronous creation of colostomy could not be identified in the claim database, it was impossible to include it as an independent variable. However, we believe that our operational definitions could cover most of the colostomy or ileostomy formation cases because those patients usually underwent APCT again or antibiotics were prescribed longer than usual. Thank you for your comments.

3. In Table 2: Demographics and Perioperative Clinical characteristics. How did authors could identify some variables were surgical complications or risk factors of AL? For example, ischemic heart disease, ischemic stroke, etc.

Response: We agree with your opinion. In Table 3, ischemic heart disease and ischemic stroke were not surgical complications. These variables were identified in each patient within 1 year before surgery. Thank you for your comments. 

4. Table 3: Risk factors of anastomotic leak from logistic regression. Age, (years) was an independent variable by multivariate analysis but not by univariate analysis? The subgroup analysis of CCI Score should be mentioned here, and no multivariate analysis for CCI score? Protective ostomy is often performed in ultra-low anterior resection (uLAR), of which might be considered as a compounding factor but not as a risk factor. 

Response: Thank you for your comments. We constructed an initial multiple model using variables that were significant based on a significance level of 0.15 in the univariable model. Because of this, in multiple models, the number of beds was excluded.

At this time, CCI and comorbidities (Diabetes, COPD, Metastatic disease, Ischemic heart disease, Ischemic stroke) were variables with strong multicollinearity, and only one of them had to be included in the model. Therefore, the model with higher explanatory power was determined to be the final model by comparing multiple models with comorbidities excluded and multiple models with CCI excluded and comorbidities included.

We described this procedure in the statistical analysis part in the manuscript as follows: The multivariable model included factors that showed the statistical significance based on univariable analysis at the significance level of 0.15, and it was evaluated by generalized variance inflation factor(VIF) for multicollinearity and the final model was chosen based on Akaike Information Criteria(AIC). 

Furthermore, robotic-assisted surgery vs laparoscopy vs open surgery is suggested to be analyzed if this procedure is related to AL?

Response: Thank you for your comments. Robotic-assisted surgery for colorectal resection is not covered by insurance in Korea. Since we used the nationwide claim data and robot surgery is not covered by insurance, it was not included in this analysis by default. 

5. Table 4. Healthcare costs by procedures & approaches. Only open vs laparoscopic approaches? How about in the comparison with robotic-assisted surgery? The relevant information regarding robotic-assisted surgery is important in recent years.

Response: Thank you for your comments. Robotic-assisted surgery for colorectal resection is not covered by insurance in Korea. Since we used the nationwide claim data and robot surgery is not covered by insurance, it was not included in this analysis by default.

6. Table 5. Economic outcomes by the presence of anastomotic leakage. Mean LOS and Median LOS for index hospitalization, (duration) was 14.22 ± 8.36 and 12 days, respectively. In fact, it was relatively longer compared to Western countries and even longer than some Asian countries.

Response: Thank you for your comments. We agree with your opinion. It is not possible to accurately analyze the cause of hospital stay being considerably longer than in other cases. Although there is a difference in the room rate in Korea, when it is covered by insurance, the cheapest room rate is about 10 dollars a day. Therefore, patients want to stay in the hospital as long as possible, rather than being discharged early. This is thought to be one of the main causes. But, over the last 5-6 years, each hospital has been applying ERAS, etc. to discharge patients as early as possible to improve the management of the hospital, so the length of stay has been shortened. Therefore, it seems that the long-term hospitalization was probably reflected in the past, and a relatively large number of patients who underwent open surgery are also considered one of the reasons.

7. If authors could use Health Insurance Review and Assessment Service (HIRA) archives claim data to analyze the difference of overall survival between AL vs non-AL patients?

Response: As the HIRA database includes only the information on reimbursement, there is no information on death, so we could not identify the mortality. Thank you for your comments.

8. In Abstract section: Male gender, comorbidities, protective ostomy, and multiple linear stapler use were associated with a higher odds of AL. The above statement should be amended according to complete results in Table 3.

Response: Thank you for your comments. As mentioned in the previous section, CCI and the rest of the comorbidities had high multicollinearity, and the multiple model with comorbidities showed better explanatory power than that with CCI. Thus, we just showed the effect size of CCI in the univariable model and excluded that in the final multivariable model.

Included in the revised manuscript

Male gender, comorbidities (diabetes, metastatic disease, ischemic heart disease, ischemic stroke), protective ostomy, and multiple linear stapler use, blood transfusion, and urinary tract injury were associated with the higher odds of AL. Older age, rectosigmoid junction cancer, AR, LAR, and laparoscopic approach were related with the reduced odds of AL.

Minor Essential Revisions:

1. Please correct the typos and grammatical error by English-editing with the certificate enclosed.

Response: Thank you for your comments. We made corrections to the manuscript accordingly. 

2. Abbreviations in the Tables must be shown their corresponding full name in the footnotes.

Response: Thank you for your comments. We made corrections to the manuscript accordingly.

6. PLOS authors have the option to publish the peer review history of their article (what does this mean?). If published, this will include your full peer review and any attached files.

Do you want your identity to be public for this peer review? For information about this choice, including consent withdrawal, please see our Privacy Policy.

 Reviewer #1: No

Reviewer #2: No

Reviewer #3: No

---

## [Editor Report · Decision Letter 1]

20 Apr 2022

Risk factors and economic burden of postoperative anastomotic leakage related events in patients who underwent surgeries for colorectal cancer

PONE-D-21-34976R1

Dear Dr. LEE,

We’re pleased to inform you that your manuscript has been judged scientifically suitable for publication and will be formally accepted for publication once it meets all outstanding technical requirements.

Kind regards,

Alberto Meyer, MD, PhD

Academic Editor

PLOS ONE

Additional Editor Comments (optional):

Dear Authors,

Thank you for accepting our recommendations for revision and incorporating the relevant changes.

I am satisfied with your response and thus happy to recommend in favour of publication of your study.

Kind regards
---

## [Editor Report · Acceptance letter]

10 May 2022

PONE-D-21-34976R1 

Risk factors and economic burden of postoperative anastomotic leakage related events in patients who underwent surgeries for colorectal cancer 

Dear Dr. Lee:

I'm pleased to inform you that your manuscript has been deemed suitable for publication in PLOS ONE. Congratulations! Your manuscript is now with our production department. 

Kind regards, 

on behalf of

Professor Alberto Meyer 

Academic Editor

PLOS ONE